# Spatiotemporal Pattern Evolution and Driving Factors of Brucellosis in China, 2003–2019

**DOI:** 10.3390/ijerph191610082

**Published:** 2022-08-15

**Authors:** Li Xu, Yijia Deng

**Affiliations:** Department of Statistics, School of Mathematics and Statistics, Guangdong University of Foreign Studies, Guangzhou 510006, China

**Keywords:** Brucellosis, spatiotemporal pattern, spatial heterogeneity, spatiotemporal scan, GeoDetector

## Abstract

Brucellosis is a prevalent zoonotic disease worldwide. However, the spatiotemporal patterns evolution and its driving factors of Brucellosis have not been well explored. In this study, spatiotemporal scan statistics were applied to describe the spatiotemporal pattern of evolution in Brucellosis from 2003 to 2019 in mainland China, and GeoDetector analysis was further conducted to explore the driving effects of environmental, meteorological, and socioeconomic factors. We identified a distinct seasonal pattern for Brucellosis, with a peak in May and lowest incidence between September and December. High-risk clusters were first observed in the northwestern pastoral areas and later expanded to the southern urban areas. The spatiotemporal heterogeneity was mainly explained by total SO_2_ emissions, average annual temperature, sheep output, and consumption of meat per capita with explanatory powers of 45.38%, 44.60%, 40.76%, and 30.46% respectively. However, the explanatory power changed over time. Specifically, the explanatory power of average annual temperature tended to decrease over time, while consumption of meat per capita and total output of animal husbandry tended to increase. The most favorable conditions for the spread of Brucellosis include 0.66–0.70 million tons of SO_2_ emissions, 9.54–11.68 °C of average annual temperature, 63.28–72.40 million heads of sheep output, and 16.81–20.58 kg consumption of meat per capita. Brucellosis remains more prevalent in traditional pastoral areas in Northwest China, with the tendency of spreading from pastoral to non-pastoral, and rural to urban, areas. Total SO_2_ emission, average annual temperature, sheep output, and consumption of meat per capita dominated the spatial heterogeneity of Brucellosis with changes in explanatory power over time.

## 1. Background

Brucellosis (human Brucellosis), caused by the *Brucella* species, is one of the most prevalent zoonotic diseases worldwide. It is transmitted through direct contact with cattle or infected animals and their product or fomites, such as undercooked meat, unpasteurized milk, and infected secretions [1]. Brucellosis causes chronic debilitating infections, with systemic symptoms (e.g., fever, headache, chills) and localized secondary infections (arthritis, endocarditis, meningitis, etc.) [2]. Due to these non-specific symptoms, Brucellosis has become one of the most commonly neglected bacterial diseases in the world [3]. According to the World Health Organization (WHO), Brucellosis has spread over 170 regions/countries with approximately 500,000 new cases being diagnosed every year, causing a widespread public health concern [4]. In China, Brucellosis was prevalent during the 1950s–1970s and then later brought under control in the 1990s due to effective prevention and control measures [3]. However, in recent years, there has been an epidemic outbreak, with a 7.8% annual increase in the number of reported cases [5]. This suggests that this new epidemic exhibits novel epidemiological characteristics [6] and that the current prevention and control measures may need be improved accordingly [5]. Additionally, the observed spatial agglomeration suggest that the epidemic may be closely associated with area-specific factors [7]. Therefore, additional focus is needed to better understand the epidemiology of the new emerging Brucellosis and to develop effective prevention and control strategies.

Previous studies have mainly focused on immunology [8], pathogenesis [9], immunological preparation development [10], or clinical diagnosis [11]. In recent years, there has been increased interest in exploring the spatiotemporal distribution of Brucellosis and its driving factors. Brucellosis hot spots were observed in pastoral areas, such as the southeastern/northern part of Brazil and the northern part of China [12,13]. Moreover, we found that Brucellosis was more prevalent in spring and much less prevalent in winter [14]. Although Brucellosis has been prevalent for a long time, the mechanisms for its spatiotemporal patterns and its evolution have not been well explored. Although some reports have revealed its mode of transmission from a geographical perspective [15,16], few studies have investigated the spatiotemporal evolution and its driving factors. Additionally, previous studies have focused on some areas with high Brucellosis incidences in China, such as Inner Mongolia [13] and Ningxia [17]. In fact, apart from the traditional pastoral areas, Brucellosis transmission has also expanded to non-pastoral areas in China in recent years. Further, most studies have only explored the individual roles of socioeconomic, environmental, or meteorological factors for the spread of Brucellosis [18,19], without considering the interplay among the different factors. Finally, previous studies have usually applied traditional methods, such as Spearman Correlation Analysis [20], Poisson Regression [18], or the Generalized Linear Model [21], to investigate associations between Brucellosis incidence and potential risk factors without utilizing the underlying spatial or temporal information. In contrast, spatial-temporal scan statistics is powerful for depicting spatiotemporal distribution [22], and the GeoDetector is helpful in investigating the driving factors of the spatiotemporal patterns of Brucellosis [23].

To address the above oversights, our study firstly conducted a spatiotemporal scan analysis to visualize the spatiotemporal pattern and its evolution of Brucellosis in mainland China during 2003–2019 and then employed the GeoDetector method to further quantify the driving effects of environmental, meteorological, and socioeconomic factors. Our study is conducted at a provincial scale and is expected to better elucidate the mechanism of Brucellosis transmission and provide evidence for the development of the strategies and measures for Brucellosis prevention and control.

## 2. Materials and Methods

### 2.1. Data and Sources

The study areas are at the municipal-level in mainland China, excluding Hong Kong, Macao, and Taiwan. The monthly and annual reported cases, incidence rates, and patient information, including gender and location (latitude and longitude), between 2003 and 2019 were extracted from the Data-Center for Chinese Public Health Science https://www.phsciencedata.cn/Share/index.jsp (accessed on 1 July 2021). Brucellosis cases were defined on the basis of epidemiologic, clinical, and laboratory criteria based on “Diagnostic criteria for Brucellosis” (2007) published by Ministry of Health [24]. All environmental data were obtained from the National Data conducted by Chinese National Bureau of Statistics http://data.stats.gov.cn (accessed on 1 July 2021), including total emissions of sulfur dioxide (SO_2_) and wastewater discharge (WWD) [25]. The meteorological data were obtained from the China Meteorological Data Sharing Service System, including average annual temperature (AAT) and average annual precipitation (AAP) [26] (Table 1). In addition, all socioeconomic data were collected from the Statistical Yearbook of each province and from the Chinese Health Statistics Yearbook, including consumption of meat per capita (consumption of pigs, cattle and sheep per capita) (CM), total output of animal husbandry (TAH) [19], consumption of milk per capita (COM) [27], sheep output (SOP), and cattle output (COP) [19]. An electronic map of China (1:4,000,000 scales) was downloaded from the National Fundamental Geographic Information System of China http://bzdt.ch.mnr.gov.cn/index.htm (accessed on 1 August 2021).

### 2.2. Statistical Methods

#### 2.2.1. Spatiotemporal Scan Statistic

We employed a Poisson retrospective spatiotemporal scan statistic by using SaTScan^TM^ to identify spatiotemporal clusters for Brucellosis [28]. We identified the most likely clusters out of a number of cylindrical candidate clusters. The space–time cylinders were determined by a scanning window, with base and height representing space and time, respectively. The scanning window moves both in space and time to cover every possible geographic location in each possible time interval. The model parameters for maximum window sizes in space and time were set to include 30% of the at-risk population and 50% of the study period, respectively [28,29].

For each window, we first calculated the expected and observed numbers of cases and then constructed a Log likelihood ratio statistic to evaluate whether the cylinder contains a cluster or not. Subsequently, we obtained the *P*-value for the detected clusters using Monte Carlo simulations. We then calculated the likelihood value as follows [22]:(1)$$\mathrm{LLR}=\log{\left(\frac{c}{E\left[c\right]}\right)}^{c}\times {\left(\frac{C-c}{C-E\left[c\right]}\right)}^{C-c}\times I$$
where *c* and *C-c* denote the observed number of cases inside and outside each window, respectively, whereas *E*[*c*] and *C-E*(*c*) represent the expected number of cases inside and outside window, respectively. Furthermore, I represents an indicator function that equals to 1 when the observed cases in the inside window are higher than the expected cases.

To ensure adequate precision, we set Monte Carlo simulations at 999 for significance testing. The scanning window with maximum statistically significant Log likelihood was identified as the most likely cluster and the other windows are ranked by value of their Log likelihood. For all analyses, we only reported the most and secondary likely clusters with statistical significance of *p* < 0.05.

#### 2.2.2. GeoDetector Method

GeoDetector is a statistical method to measure the degree of spatial stratified heterogeneity, as well as being a tool to analyze differences and similarities of intra-layer and inter-layer variance, including Factor Detector, Interaction Detection, Risk Detector, and ecological detector [23].

##### Factor Detector

We performed Factor Detector analyses to detect the spatial heterogeneity of Brucellosis incidence and to calculate the explanatory power of its potential driving factors. This is expressed by a *q* value, which can quantify the effects of potential risk factors responsible for the spatial heterogeneity of Brucellosis in mainland China during 2003–2019. The *q* statistic value was calculated as follows:(2)$$q=1-\frac{1}{N\sigma^{2}}\sum_{h=1}^{L}N_{h}\sigma_{h}^{2}$$
where *q* represents the explanatory power of factor (*X*) on the spatial heterogeneity of Brucellosis cases (*Y*). The value of *q* is required to be within [0, 1]; *h =* 1,..., L, which denotes the stratification of *X* and *Y.* Further, *N_h_* and *N* are the sample sizes for the stratum *h* and all units; $\sigma_{h}^{2}$ and $\sigma^{2}$ are the variances of Y for the layer *h* and the whole study area. The larger the *q* value is, the stronger the explanation power on spatial heterogeneity. Though q value has not been clearly defined [23], most researchers believe that a q value above 0.2–0.5 indicates a strong explanatory power [30,31,32]. In our study, factors with a q value higher than 0.3 and the corresponding *p* value less than 0.05 would be identified as a dominating factor on the spatial heterogeneity of Brucellosis.

##### Interaction Detector

We further conducted Interaction Detector analyses to determine whether the interactions among significant risk factors would increase their single explanatory power. The relationship between factors is categorized in Table 2.

##### Risk Detector

Finally, we performed Risk Detector analyses to explore the most favorable living conditions for Brucellosis. We divided the study area into several sub-regions according to the classification of each potential risk factor, then compared the differences in the average incidence between the sub-regions using a *t*-test. The t-statistic was calculated as follows:(3)$$t_{{\bar{Y}}_{h=1}-{\bar{Y}}_{h=2}}=\frac{{\bar{Y}}_{h=1}-{\bar{Y}}_{h=2}}{{\left[\frac{var\left({\bar{Y}}_{h=1}\right)}{N_{h=1}}+\frac{var\left({\bar{Y}}_{h=2}\right)}{N_{h=2}}\right]}^{1/2}}$$
where $\bar{Y}$*_h_* denotes the average incidence of Brucellosis in stratum *h*, *N_h_* is the sample size of stratum *h*, and *Var* represents the variance. Sub-regions with a greater significance have a higher risk of Brucellosis.

## 3. Results 

### 3.1. Trend Analysis of Brucellosis Epidemiological Data

The temporal trend of the 2003–2019 Brucellosis outbreak in mainland China is shown in Figure 1. Gender information of patients was only available during 2004–2010. In total, 162,329 cases were reported during 2004–2010. Among them, 121,081 (74.59%) were males and 41,248 (25.41%) were females, with a male-to-female ratio of (2.94:1). The incidence in males was higher than its counterpart significantly (χ^2^ = 130.65, *p* = 0.00). According to its epidemic characteristics [1], the study period was further divided into three phases with significant difference in the number of cases (χ^2^ = 89.97, *p* < 0.05). Phase 1 (2003–2014) was the outbreak period with the number of new cases rising from 11,800 to 57,613 and peaking in 2014. Phase 2 (2014–2017) was the mild period with the number of new cases decreasing from 57,613 to 28,790, indicating some level of improved control measures. Phases 3 (2017–2019) was the recurrence period with the number of new cases increasing to 44,134, indicating a rebound outbreak of the epidemic. Moreover, the Brucellosis outbreak exhibited an obvious seasonal pattern. Specifically, a higher incidence was observed between April and July, with a peak in May, while a lower incidence was observed between September and December (Figure 2, Monthly incidence of Brucellosis in China during 2016–2017).

### 3.2. Spatiotemporal Pattern of Brucellosis

#### 3.2.1. Spatiotemporal Pattern of Brucellosis in the Outbreak Period

We identified four statistically significant spatiotemporal clusters, including one most likely cluster and three secondary clusters, covering fifteen provinces in total (Figure 3a). The most likely cluster was identified between January 2009 and December 2014, centering on Inner Mongolia and expanding to Shanxi province, with a radius of 335.33 km. The first of the secondary clusters appeared at the same time as the most likely cluster but centered on Jilin province and expanded to Heilongjiang province, with a radius of 241.64 km. The second of the secondary clusters was identified between January 2013 and December 2014, centering on Shandong province and expanding to Hebei province, with a radius of 268.15 km. The third of the secondary clusters appeared between January 2014 and December 2014 and centered on Tibet and expanded to Sichuan, Yunnan, Gansu, Chongqing, Qinghai, Guizhou, and Ningxia provinces, with a radius of 1697.78 km. During the outbreak period, Brucellosis was most prevalent in the traditional pastoral areas such as Inner Mongolia, but it tended to expand to the northern areas of China (e.g., Jilin and Heilongjiang province) and also expanded from the northern traditional pastoral areas to central urban areas.

#### 3.2.2. Spatiotemporal Pattern of Brucellosis in the Mild Period

We identified two statistically significant spatiotemporal clusters, including one most likely cluster and one secondary cluster, covering eight provinces in total (Figure 3b). The most likely cluster was identified between January 2014 and December 2015, centering on the northwest of China and expanding to Xinjiang, Qinghai, Tibet, Gansu, Ningxia, and Inner Mongolia, with a radius of 2000.09 km. The secondary cluster appeared at the same time as the most likely cluster but covered Shanxi and Hebei, with a radius of 173.58 km. During the mild period, Brucellosis was more prevalent in northwestern China.

#### 3.2.3. Spatiotemporal Pattern of Brucellosis in the Recurrence Period

We identified three statistically significant clusters, including one most likely cluster and two secondary clusters, covering eleven provinces in total (Figure 3c). The most likely cluster was identified in 2019, centering on Xinjiang Province and expanding to Qinghai, Tibet, Gansu, Ningxia, Inner Mongolia, and Sichuan, with a radius of 2000.09 km. The first of the secondary clusters appeared in 2018, centering on Heilongjiang, with a radius of 152.47 km. The second of the secondary clusters was also identified in 2019, centering on Shanxi and expanding to Hebe and Henan, with a radius of 198.62 km. During the recurrence period, Brucellosis reclustered in northern China, followed by several new clusters in Hebei, Henan, and Sichuan, showing a trend of expanding to the southern areas.

Characteristics of spatiotemporal clusters in each period are summarized in Table 3. Clearly, Brucellosis was still centered in traditional pastoral areas (i.e., Xinjiang). The radius of the clusters increased steadily, from 335.33 km to 2000.09 km. In addition, coverage of clusters decreased from 15 provinces in the outbreak period to 8 provinces in the mild period, followed by a subsequent increase to 11 provinces in the recurrence period. To conclude, the epidemic was still serious in traditional pastoral areas (Figure 4, Major pastoral areas in China) (four major pastoral areas in China are Inner Mongolia, Xinjiang, Tibet, and Qinghai according to Ministry of Agriculture of China), while it was gradually spreading from pastoral to non-pastoral areas (i.e., rural to urban areas).

### 3.3. Driving Factors for Spatial Heterogeneity of Brucellosis

#### 3.3.1. Driving Effects of Potential Risk Factors on the Spatial Heterogeneity of Brucellosis

We found significant associations between the spatial heterogeneity of Brucellosis and total SO_2_ emissions, average annual temperature, sheep output, consumption of meat per capita, total output of animal husbandry, average annual precipitation, consumption of milk per capita, cattle output, and wastewater discharge. The explanatory powers of these factors were 45.38%, 44.60%, 40.76%, 30.46%, 28.12%, 20.72%, 20.30%, 20.11%, and 15.30%, respectively. These findings suggested that the first four factors dominated the spatial heterogeneity of Brucellosis, especially for total SO_2_ emissions. (For more information about dominating factor, please see Section of *Factor Detector*, page 4, from line 150 to line 152.) 

Factor Detector analysis for each period showed that the explanatory power of these factors changed over time. In the outbreak period, the sheep output, total SO_2_ emissions, and average annual temperature dominated the spatial heterogeneity of Brucellosis, with explanatory powers of 40.40%, 39.29%, and 35.13%, respectively. In the mild period, the explanatory power of total SO_2_ emissions decreased to 34.94%, followed by sheep output, consumption of meat per capita, and average annual temperature, with the corresponding values of 32.84%, 30.26%, and 30.09%, respectively. During the recurrence period, the explanatory power for consumption of meat per capita increased dramatically to 49.65%, followed by total SO_2_ emission (47.18%), sheep output (42.43%), and total output of animal husbandry (35.47%).

A detailed list of the explanatory power for each factor in each period is shown in Table 4. Although fluctuating, the explanatory power of total SO_2_ emissions and sheep output always ranked in the top three, indicating their dominant role in the spatial heterogeneity of Brucellosis. Different from that of total SO_2_ emissions and sheep output, the explanatory power of average annual temperature gradually declined, illustrating its significant yet weakening effects. In contrast, there was an upward trend for the consumption of meat per capita and total output of animal husbandry, suggesting their increasing impact on the spatial heterogeneity of Brucellosis. Additionally, the explanatory power of average annual precipitation, consumption of milk per capita and cattle output remained around 20% while that of wastewater discharge accounted for about 13%. In conclusion, environmental, meteorological, and socioeconomic factors all played an important role in spatial heterogeneity of Brucellosis. Specifically, the driving effects of environmental and meteorological factors were relatively stable, while that of socioeconomic factors exhibited an increasing trend.

#### 3.3.2. Analysis of Influence of Consumption and Production on Brucellosis Clusters (Areas in Most Likely Cluster in Recurrence Period)

Table 4 shows that the explanatory power of most socioeconomic factors exhibited an increasing trend in recurrence period, with an exception for cattle output. This finding indicated that there may be close relationship between the recurrence of Brucellosis and the varying of socioeconomic factors. Thus, we conducted a deeper analysis with emphasis on the impact of consumption and production on Brucellosis (Table 5).

Table 5 showed that: (1) The consumption of pigs and cattle dominated consumption while the output of sheep dominated production for Qinghai, which indicated that consumption of Brucella-contaminated pigs and beef and direct contact with infected sheep might have a major influence on the recurrence of Brucellosis in Qinhai. (2) The consumption of beef and output of sheep dominated consumption and production, respectively, for Tibet, indicating that the consumption of Brucella-contaminated beef and direct contact with infected sheep may be a major source for the recurrence of Brucellosis in Tibet. (3) The consumption of pigs and output of sheep dominated consumption and production, respectively, for Gansu, Ningxia, and Inner Mongolia. This finding indicated that people living in these areas mainly get infected through the consumption of Brucella-contaminated pigs and direct contact with infected sheep in the recurrence period. (4) Both the consumption and production of pigs dominated in Sichuan, indicating that the consumption of Brucella-contaminated pigs and direct contact with infected pigs were major sources of recurrence of Brucellosis for people living in Sichuan. In conclusion, the frequent consumption of Brucella-infected pigs and direct contact with infected sheep were main causes of Brucellosis in the recurrence period for Brucellosis clusters in China.

#### 3.3.3. Interactions among Driving Factors

The interactions among any two factors enhanced the explanatory power for any single factor (Figure 5, Interaction hotspot maps for factors influencing Brucellosis incidence, 2003–2019). For example, the *q* values for total SO_2_ emissions and average annual temperature were 0.39 and 0.45, respectively, yet this value increased to 0.76 after accounting for their interaction. The same held true for all pairwise interactions among these factors.

#### 3.3.4. Favorable Conditions for Spread of Brucellosis

Favorable conditions for the spread of Brucellosis in the whole study period and each period are shown in Table 6. During the whole study period, areas with high prevalence of Brucellosis exhibited the following characteristics: 0.66–0.70 million tons of SO_2_ emissions, 2444.04–3245.28 million tons of wastewater discharge,9.54–11.68 °C of average annual temperature, 421.54–630.76 mm of average annual precipitation, 16.81–20.58 kg consumption of meat per capita, 872.93–1256.43 billion yuan of total output of animal husbandry, 12.99–17.26 kg consumption of milk per capita, 63.28–72.40 million heads of sheep output, and 2.93–3.51 million heads of cattle output. The results of each period were basically consistent with that of the whole study period, with a slight difference in the total SO_2_ emissions (0.73–0.78 million tons in the recurrence period period), wastewater discharge (1065.10–207,136.33 million tons in the mild period), average annual temperature (7.41–9.589 °C in the recurrence period), total output of animal husbandry (1828.70–2548.30 billion yuan in the mild period), consumption of milk per capita (17.52–20.79 in the recurrence period), and cattle output (2.09–2.61 in the mild period).

## 4. Discussion

In recent years, Brucellosis has become a serious global health problem with a considerable disease burden on human health and livestock productivity. However, the spatiotemporal heterogeneity and driving factors behind Brucellosis transmission in China remain unclear. In this study, we investigated the evolution of spatiotemporal patterns of Brucellosis in China between 2003 and 2019 and explored the potential driving factors from environmental, meteorological, and socioeconomic perspectives by conducting spatiotemporal scan statistics and GeoDetector analysis. We observed obvious seasonal and regional heterogeneity for Brucellosis incidence and quantified effects of driving factors that changed over time. Furthermore, we calculated the most favorable conditions for Brucellosis transmission in China.

There was a significant spatiotemporal heterogeneity for Brucellosis in China between 2003 and 2019. The epidemic in our study period was further divided into the outbreak period, mild period, and recurrence period, consistent with the features of Category B infectious diseases (e.g., COVID-19 and SARS) [33,34]. It may indicate that the methods applied in our study is also applicable for exploring the spatiotemporal heterogeneity and the corresponding driving factors of SARS or COVID-19. In addition, high-risk areas continued to cluster in traditional pastoral areas such as Inner Mongolia, Xinjiang, Qinghai, and Tibet, consistent with previous findings [5]. Since 2014, however, several new clusters have gradually emerged in non-pastoral areas including Henan and Sichuan province, with a tendency of shifting from northwest inland to the southern areas and from pastoral areas to urban areas in China. 

Brucellosis infection is complex and involves natural, individual, and societal factors. Natural factors such as temperature, precipitation, and wastewater discharge have a direct effect on the living environment of Brucellosis [17,35]. Individual occupational characteristics, education level, or the frequency of exposure to dairy products also played vital roles [16,18,36]. Last but not least, economic development may also affect the frequency of human contact with *Brucella*, thereby contributing to the epidemic [7,18,37]. Our study demonstrated that the spatial heterogeneity of Brucellosis during the study period was driven by various factors, including total SO_2_ emissions, average annual temperature, sheep output, consumption of meat per capita, total output of animal husbandry, average annual precipitation, consumption of milk per capita, cattle output, and wastewater discharge. Interactions between these factors would increase Brucellosis transmission, which agrees with previous studies [38].

Environmental factors (including total SO_2_ emissions and wastewater discharge) were one of the most important driving factors, especially for total SO_2_ emissions in the recurrence period (2017–2019). The potential reasons for this finding are complex. One possible explanation is that the rapid development of industrialization in China contributed a lot to higher SO_2_ emissions, which is the main component of acid rain and may potentially inhibit the spread of *Brucella* [17,39]. Yet, on the other hand, higher SO_2_ emissions could also reflect the changes of living habits in the populations [40]. This possibly reflects increasing demand for pigs, cattle, sheep, and dairy products, which would lead to a higher risk of Brucellosis transmission. So far, the relationship between SO_2_ emissions and spatial heterogeneity of Brucellosis has not been fully explored. Therefore, the exact mechanism of SO_2_ emission in the spatial heterogeneity of Brucellosis remains elusive and warrants more studies. Moreover, wastewater discharge played a weak but stable role in Brucellosis spatial heterogeneity due to the fact that open water sources may also be a source of infection when contaminated with wastewater from farms [41]. In a word, our study indicated the important role of environmental change on the reemergence of infectious disease (such as Brucellosis). In fact, a warming and unstable climate is becoming more and more important in the global emergence and resurgence of infectious diseases, such as tuberculosis and diphtheria [42,43]. It is necessary to mention that though environmental pollution has been proven to be a main cause of Brucellosis spatial heterogeneity [37], no study has demonstrated SO_2_ to be a dominating factor. This inconsistency could be partly explained by the fact that previous studies have only focused on a certain region and employed traditional statistical methods without utilizing the underlying spatiotemporal information. In contrast, we conducted analyses from the national perspective and applied geographic detectors to better reveal the association between the total SO_2_ emissions and spatial heterogeneity of Brucellosis [23].

Meteorological factors (including average annual temperature and average annual precipitation) also had an essential impact on the spatial heterogeneity of Brucellosis, especially for average annual temperature. Our findings are consistent with previous findings that low temperatures and high humidity are as conducive to the long-term survival of *Brucella* as many other pathogens in the environment [6]. This could be explained with that low temperatures and high humidity may increase the risk of exposure to this pathogen [35]. Nevertheless, the impact of average annual temperature has declined since 2014. The underlying reason for this finding remains unclear but perhaps is due to an increase in the relative explanatory power for total SO_2_ emissions, consumption of meat per capita, total output of animal husbandry, etc.

Last but not least, socioeconomic factors, including sheep output, cattle output, the consumption of meat per capita, total output of animal husbandry, and consumption of milk per capita, were other important driving factors contributing to the spatial heterogeneity of Brucellosis. Firstly, sheep output, instead of cattle output, was found to be a key driving factor. A possible explanation may be that B. melitensisis is the main pathogen in China [44]. Herders get infected through direct contact with Brucella-infected sheep or their products [45]. Therefore, Brucellosis tends to occur in the northern and north-western areas with large scales of sheep, and control measures are strongly recommended to target those areas where there is exposure to sheep or goats. Additionally, a trend of increasing impact was observed in consumption of meat per capita and total output of animal husbandry, especially in the recurrence period. Among them, the former was the most important factor. The potential reason may be that a greater consumption of meat is associated with a higher consumption of contaminated meat [21]. Particularly, it has been found that the frequent consumption of Brucella-infected pigs and direct contact with infected sheep could be main causes in China in recurrence periods, which is consistent with the previous study [4,44]. Moreover, the upward change in that of total output husbandry may be explained by more food circulation and consumption from infected animals [7]. In a word, the increasing effect of consumption of meat per capita and total output of animal husbandry probably suggest more food-borne cases in the future. The consecutively reported cases of food-borne Brucellosis infection in urban cities of China in 2019–2021 also supported our finding [5,46]. Finally, though fluctuating, the effect of consumption of milk per capita remained relatively robust, which might indicate the important role of unpasteurized milk and milk products in the transmission of Brucellosis from animals to humans [47].

Our study contributes to the present research in the following ways: (1) Our study was conducted at the provincial level to investigate the spatiotemporal pattern. This may better reveal the outbreak and the evolutionary features of Brucellosis in China from a holistic view. (2) Our study focused on spatiotemporal patterns and Brucellosis evolution in China, which can provide an in-depth understanding for the dynamic spread of Brucellosis. (3) Our study explored and quantified the effects that drove the spatiotemporal heterogeneity of Brucellosis from the ecological–climate–socioeconomic interface, which may help us better understand the epidemiology of Brucellosis.

Several limitations of this study merit discussion. Firstly, the reported cases in our study may be underestimated as some patients in China were not diagnosed due to the non-specific nature of Brucellosis symptoms. However, we obtained reported cases from Chinese Public Health Science Data Center, which is controlled by the Chinese Center for Disease Control and Prevention. Thus, we believe that the main results in our study are reliable. Additionally, although we tried to explain the driving effects for spatial heterogeneity of Brucellosis from an ecological, climate, and socio-economic perspective, socioeconomic data concerning the infected populations (e.g., age, occupation, rural or urban life, the proximity of cattle farms to the place of housing, consumption habits of meat and dairy products, etc.) are also an important perspective and should not be ignored. However, factors about human behavior are not available at present. Therefore, further study is needed if data are to be available to incorporate various factors involving nature, society, and individual behavior to investigate the transmission mechanisms of Brucellosis in depth.

## 5. Conclusions

We found a significant spatiotemporal heterogeneity of Brucellosis in China during 2003–2019. In terms of temporal heterogeneity, Brucellosis was more prevalent in spring but much less prevalent in winter. Regarding spatial heterogeneity, it remained more prevalent in traditional pastoral areas in Northwest China while gradually spreading from pastoral to non-pastoral areas and from rural to urban areas.

Total SO_2_ emissions, average annual temperature, sheep output, and consumption of meat per capita dominated the spatial heterogeneity of Brucellosis with the effect of socioeconomic factors increasing over time. Moreover, pairwise interactions enhanced the driving effects for all significant driving factors. Finally, we identified favorable conditions for the spread of Brucellosis. To conclude, promptly adjusting and improving the prevention and control strategies for Brucellosis is strongly recommended, especially for the traditional pastoral areas in the northwestern part of China and urban areas where food-borne cases have been reported.

## Figures and Tables

**Figure 1 ijerph-19-10082-f001:**
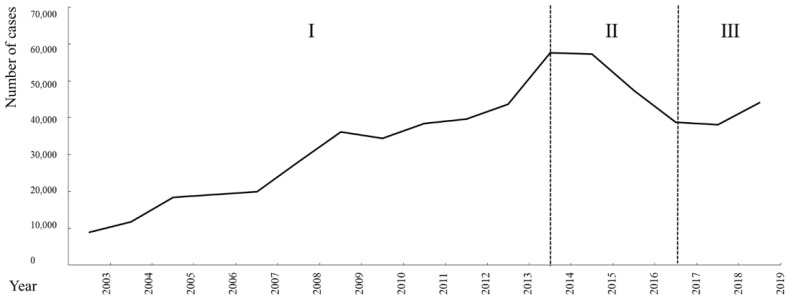
Number of new cases of Brucellosis in China during 2003–2019.

**Figure 2 ijerph-19-10082-f002:**
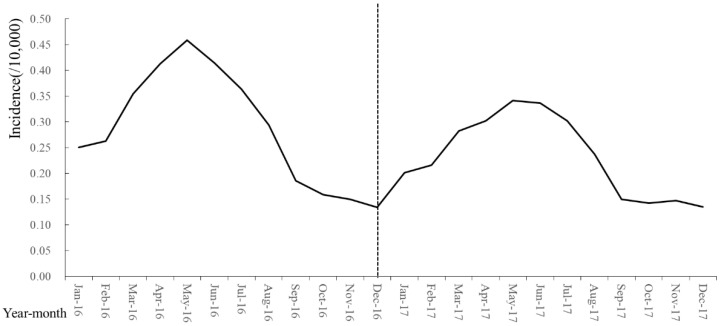
Monthly incidence of Brucellosis in China during 2016–2017. (The monthly incidence of patients was only available during 2016–2017.)

**Figure 3 ijerph-19-10082-f003:**
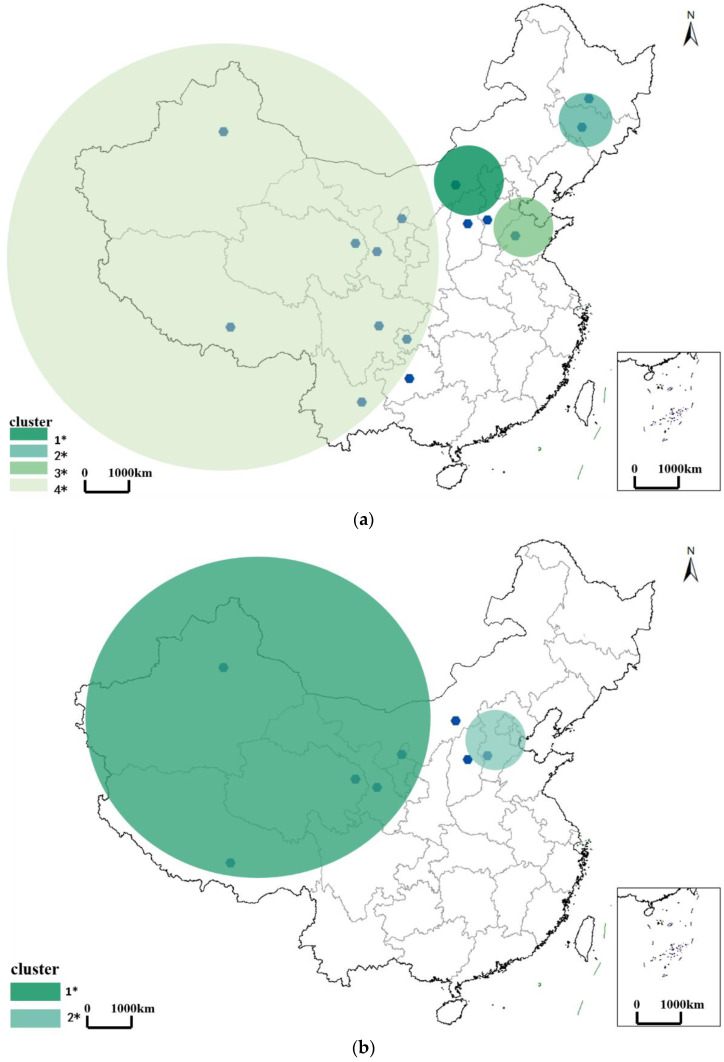
Spatiotemporal pattern of Brucellosis in: (**a**): outbreak period (2003–2014); (**b**) mild period (2014–2017); (**c**) recurrence period (2017–2019). * 1: most likely cluster; 2: the secondary cluster Ⅰ; 3: the secondary cluster Ⅱ; 4: the secondary cluster Ⅲ.

**Figure 4 ijerph-19-10082-f004:**
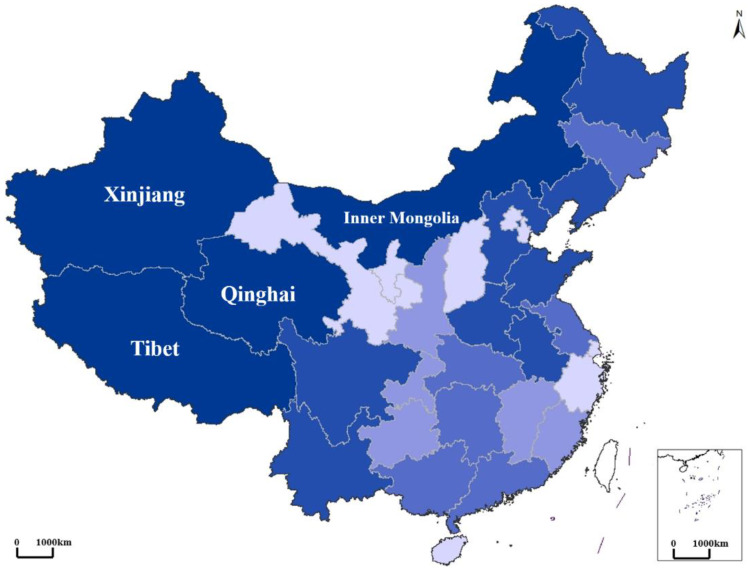
Major pastoral areas in China (based on grassland area, livestock volume, and beef and sheep output).

**Figure 5 ijerph-19-10082-f005:**
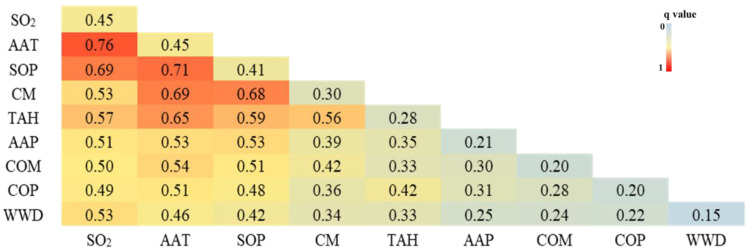
Interaction hotspot maps for factors influencing Brucellosis incidence, 2003–2019. SO_2:_ total emissions of sulfur dioxide; AAT: average annual temperature; SOP: sheep output; CM: consumption of meat per capita; TAH: total output of animal husbandry; AAP: average annual precipitation; COM: consumption of milk per capita; COP: cattle output; WWD: wastewater discharge.

**Table 1 ijerph-19-10082-t001:** Description of variables.

Type of Factor	Detection Factor	Measurement Units	Range of Time	Data Source
Environment	SO_2_	tons (t)	2003–2019	Statistical Yearbook of each province in China
	WWD	tons (t)		
Meteorology	AAT	Celsius (°C)	2003–2019	China Meteorological Data Sharing Service System
	AAP	mm		
Socioeconomic	CM	kg	2003–2019	Statistical Yearbook of each province in China
	TAH	Yuan		
	COM	kg		Chinese Health Statistics Yearbook
	SOP	heads		Statistical Yearbook of each province in China
	COP	heads		

SO_2:_ total emissions of sulfur dioxide; WWD: wastewater discharge; AAT: average annual temperature; AAP: average annual precipitation; CM: consumption of meat per capita; TAH: total output of animal husbandry; COM: consumption of milk per capita; SOP: sheep output; COP: cattle output.

**Table 2 ijerph-19-10082-t002:** Types of interaction between two factors.

Description	Interaction
q(*X*1∩*X*2) < Min(q(*X*1), q(*X*2))	Weakened, Nonlinear
q(*X*1∩*X*2) = q(*X*1) + q(*X*2)	Independent
q(*X*1∩*X*2) > Max(q(*X*1) + q(*X*2))	Enhanced, Double factors

**Table 3 ijerph-19-10082-t003:** Characteristics of statistically significant spatiotemporal clusters of Brucellosis.

	Center *	Radius **	Areas ***	Number of Provinces
Outbreak period	Inner Mongolia	335.33 km	Inner Mongolia, Shanxi	15
Mild period	Xinjiang	2000.09 km	Xinjiang, Qinghai, Tibet, Gansu, Ningxia, and Inner Mongolia	8
Recurrence period	Xinjiang	2000.09 km	Qinghai, Tibet, Gansu, Ningxia, Inner Mongolia, and Sichuan	11

* center of most likely cluster. ** radius of most likely cluster. *** areas in most likely cluster.

**Table 4 ijerph-19-10082-t004:** The explanatory power of the influencing factors on spatial heterogeneity of Brucellosis (%).

	SO_2_	AAT	SOP	CM	TAH	AAP	COM	COP	WWD
2003–2019	**45.38**	**44.6**	**40.76**	**30.46**	28.12	20.72	20.30	20.11	15.3
Outbreak period	**39.29**	**35.13**	**40.4**	27.98	24.66	20.79	19.05	21.36	12.05
Mild period	**34.94**	**30.09**	**32.84**	**30.26**	29.81	25.26	18.29	22.43	12.36
Recurrence period	**47.18**	28.9	**42.43**	**49.65**	**35.47**	22.77	24.75	23.59	16.55

All factors were statistical significance at the level of 5%. SO_2_: total emissions of sulfur dioxide; AAT: average annual temperature; SOP: sheep output; CM: consumption of meat per capita; TAH: total output of animal husbandry; AAP: average annual precipitation; COM: consumption of milk per capita; COP: cattle output; WWD: wastewater discharge.

**Table 5 ijerph-19-10082-t005:** Production and consumption distribution of meat in clusters areas (recurrence period) (%).

Areas in Cluster	Consumption			Production		
	Pigs	Cattle	Sheep	Pigs	Cattle	Sheep
Qinghai	**41.12**	**32.46**	26.42	10.82	13.76	**75.42**
Tibet	23.32	**58.00**	18.68	3.15	28.47	**68.38**
Gansu	**77.92**	8.11	13.97	28.66	8.70	**62.63**
Ningxia	**43.51**	27.84	28.65	14.42	9.72	**75.86**
Inner Mongolia	**57.58**	14.05	28.36	11.09	4.83	**84.08**
Sichuan	**94.27**	4.49	1.24	**74.35**	3.50	22.15

**Table 6 ijerph-19-10082-t006:** Favorable conditions for the spread of Brucellosis.

	SO_2_	WWD	AAT	AAP	CM	TAH	COM	SOP	COP
	/Million t	/Million t	/°C	/mm	/kg	/Billion Yuan	/kg	/Million Heads	/Million Heads
2003–2019	0.66–0.70	2444.04–3245.28	9.54–11.68	421.54–630.76	16.81–20.58	872.93–1256.43	12.99–17.26	63.28–72.40	2.93–3.51
Outbreak period	0.70–0.73	2245.28–3046.52	9.59–11.77	478.94–668.97	15.58–23.12	770.90–1147.42	11.69–14.91	64.21–78.96	3.21–3.57
Mild period	0.70–0.73	1065.10–207,136.33	9.59–11.77	417.53–623.77	16.89–21.54	1828.70–2548.30	13.37–16.59	58.46–70.09	2.09–2.61
Recurrence period	0.73–0.78	2071.36–3077.63	7.41–9.589	440.60–674.70	15.47–20.42	757.20–1239.60	17.52–20.79	64.01–73.71	2.91–3.32

All factors were statistical significance at the level of 5%. SO_2:_ total emissions of sulfur dioxide; WWD: wastewater discharge; AAT: average annual temperature; AAP: average annual precipitation; CM: consumption of meat per capita; TAH: total output of animal husbandry; COM: consumption of milk per capita; SOP: sheep output; COP: cattle output.

## Data Availability

All the data used in this study are from public sources.

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
