# Peer review of "Spatiotemporal Pattern Evolution and Driving Factors of Brucellosis in China, 2003–2019"

_ijerph, 2022, doi:10.3390/ijerph191610082_

Round 1

Reviewer 1 Report

This is a very interesting study investigating driving factors potentially associated with variations in the incidence of human brucellosis in China mainland. However, I have major concerns about the design of the study.

Major comments.

The major limitation of the study concerns the choice of the indicators studied to explain variations in the geographic spread and incidence of human brucellosis. The modes of transmission of this disease are essentially linked to contact with Brucella-infected farm animals and the consumption of food from contaminated animals. There is almost now human-to-human transmission of the disease. Some major epidemiological factors potentially associated with brucellosis incidence variations include changes in the consumption of unpasteurized milk or milk products, or at least the global consumption of milk and other dairy products, the size of cattle farms, farming habits (e.g. multiplication of intensive farming), and diagnostic methods of human brucellosis. A deeper analysis of the factors involved in the spatiotemporal distribution and incidence of brucellosis cases could be obtained if demographic and socioeconomic data concerning the infected populations were available (age, sex, occupation, rural or urban life, the proximity of cattle farms to the place of housing, consumption habits of meat and dairy products, etc.). 

I’m also not convinced by the authors’ explanations for the association of specific criteria with spatiotemporal spread and incidence of human brucellosis cases.

The role of SO2 on brucellosis incidence (line 322) is unclear. Higher SO2 production likely reflects changes in industrial activities, which might be associated with changes in the populations living habits. It is unlikely that pH changes in the environment have a major impact on brucellosis incidence since this disease is primarily food-borne. Indeed, how authors can explain that brucellosis reemerged in the third period of the study while SO2 emission was decreasing?

The argument that the illegal discharge of industrial wastewater might increase brucellosis incidence (line 335) is not convincing. Could the illegal discharge of wastewater favor the environmental spread of Brucella?

The high density of indoor livestock in low temperatures (line 341) could explain a higher incidence in cattle breeders in winter. This is not compatible with the study data. Low temperatures and high humidity can favor the long-term survival of Brucella as many other pathogens in the environment. This could represent a higher risk of exposure to this pathogen in people in contact with infected herds.

More consumption of cattle (line 351) might be associated with more frequent consumption of Brucella-contaminated meat. It may also reflect higher consumption of other cattle products such as milk and other dairy products. Overall, higher consumption of contaminated food products is more likely than direct contact with infected animals during the circulation of animals. This is especially true for people living in urban areas for which brucellosis is most often a food-borne infection. These people have usually rare contact with cattle. Total output husbandry may only reflect more food production and circulation from infected cattle.

I don't understand the potential relationship between brucellosis incidence and the number of hospital beds in a specific location, since this latter criterion is likely not to vary too much over time and certainly is not dependent on brucellosis incidence. A low number of hospital beds per 1000 persons might be associated with rurality.

Minor comments

Background, first paragraph. Brucellosis may be an acute, subacute, or chronic disease, with various clinical manifestations including systemic symptoms (fever, chills) and localized secondary infections (arthritis, endocarditis, meningitis, etc.). Besides, brucellosis is also transmitted directly through contact with cattle or their fomites.

Line 240. “These findings suggested that the first three factors dominated the spatial heterogeneity of Brucellosis, especially for total SO2 emissions.” Why only these three factors can be considered the most important to explain spatial heterogeneity in the spread of brucellosis. There is likely no causal relationship, especially for SO2 production.

Figure 3. The quality of the legends must be improved. For brucellosis patterns, it would be helpful for the reader to keep the same color for the same region.

Reviewer 2 Report

Dear Dr. Elma Yu and authors, thank you for allowing me to review your manuscript «Spatiotemporal pattern evolution and driving factors of Brucellosis in China, 2003–2019».

This subject is of high importance for health sciences and public health authorities. It brings significant knowledge to literature with novel discussion, mainly in the spatial-temporal perspective for Brucellosis investigation and the inclusion of SO2 emission. 

I have some questions and recommendations.

METHODS section:

In the data and sources section, please include a few sentences describing the confirmed/non-confirmed cases according to the Chinese Public Healh system. What methods and considerations to be a case/non-case?

As in Table 1, please add the abbreviation of the variables for all tables and Figures. It will help the reader understand which variables you are talking about. instead of a, b, c… I recommend including “Abbreviation: WWD, wastewater discharge; AAT…” below the graphical/table component.

It is unclear how the authors choose the three periods: 2003-2014; 2014-2017; and 2017-2019. Is it based on the epidemiological number of cases/incidence? Please clarify.

RESULTS section:

In the subtitle “trend analysis of Brucellosis Epidemic”, would be more appropriate to use “trend analysis of Brucellosis epidemiological data” (or something similar) because the authors divided between three phases?

Please increase the scale for the number of cases and incidence in Figures 1 and 2, showing more intervals. This will allows seeing the changes/patterns better. In Figure 2, please change the numbers 1, 2, 3… to Jan., Feb., Mar., etc. For Figures 1 and 2, I suggest indicating the three Phases with some editing, transparence, or dotted lines… In figure 1, please edit the years to be in vertical (or 45o) instead of horizontal lines.

The significant extension of the clusters, for instance, 1697 km, 2000 km, brings confusion about the idea of the extension of the Chinese space and cities. For example,  the authors make assumptions about the cluster to be in the pastoral areas; however, with this size of clusters, I believe the clusters also cover big cities of these provinces, right? There is a generalization assuming the clusters in the pastoral areas, but the pattern of land use and land cover was not analyzed in the study. In the discussion section, this will be more discussed, but it seems this is not a method/result of this research.  In line 216, the authors wrote, “Clearly, is still centered in traditional pastoral areas …”  It is unclear, and I think there is no study of land use in this manuscript. Please explain everything related to “pastoral/ urban” assumptions better. Bring some data or maps to show what changes the authors know/recognize but that do not make sense for a foreign reader.

Please try to clarify the number of hospital beds variable importance. 

DISCUSSION

Line 308-310: what is common between SARS, COVID-19, and Brucellosis?

What do the authors justify Brucellosis clusters to be in the pastoral areas? Is that related to local consumption?  If this could be related, would cattle consumption per capita be more important? Is there an idea of producing and consumption? 

Perhaps one of the most interesting results of the manuscript is the success of the SO2 variable. I suggest it relate it with a more profound discussion about climate change/ environmental change and the reemergence of diseases (such as Brucellosis), which would be appropriate with the scope of the journal and a subject of high importance.

Round 2

Reviewer 1 Report

Dear authors. Thank you for considering my comments. The revised version of the manuscript is more meaningful and interesting for readers. Best regards.